biomechanics, physiology

limb joint, oscillation damping, locomotion, walking, cycling, internal work

**Author for correspondence:**
Alberto E. Minetti
e-mail: alberto.minetti@unimi.it

# Frictional internal work of damped limbs oscillation in human locomotion

Alberto E. Minetti, Alex P. Moorhead and Gaspare Pavei

Physiology Division, Department of Pathophysiology and Transplants, University of Milan, Via Mangiagalli 32, 20133 Milan, Italy

AEM, 0000-0002-0120-4406; APM, 0000-0002-8646-6420; GP, 0000-0002-0109-4964

Joint friction has never previously been considered in the computation of mechanical and metabolic energy balance of human and animal (loco)motion, which heretofore included just muscle work to move the body centre of mass (external work) and body segments with respect to it. This happened mainly because, having been previously measured *ex vivo*, friction was considered to be almost negligible. Present evidences of *in vivo* damping of limb oscillations, motion captured and processed by a suited mathematical model, show that: (a) the time course is exponential, suggesting a viscous friction operated by the all biological tissues involved; (b) during the swing phase, upper limbs report a friction close to one-sixth of the lower limbs; (c) when lower limbs are loaded, in an upside-down body posture allowing to investigate the hip joint subjected to compressive forces as during the stance phase, friction is much higher and load dependent; and (d) the friction of the four limbs during locomotion leads to an additional internal work that is a remarkable fraction of the mechanical external work. These unprecedented results redefine the partitioning of the energy balance of locomotion, the internal work components, muscle and transmission efficiency, and potentially readjust the mechanical paradigm of the different gaits.

## 1. Introduction

The assessment of total mechanical work of terrestrial locomotion is crucial in explaining metabolic economy of gaits (the cost of transport) [1,2] and in investigating the components of progression efficiency (muscle and transmission efficiencies) [3–5]. This helps both when studying fundamental paradigms and optimization of motion [6,7], and when dealing with assisted, enhanced and impaired forms of human and animal locomotion [8–11].

The (positive) external mechanical work ($W_{EXT}$) [12] necessary to raise and accelerate the body centre of mass (BCoM) sometimes incorporates work against external friction such as air drag acting on the body surface [13] and ground friction (although the latter is negligible whenever there is no apparent sliding between the point of contact and the terrain). Early biomechanists realized that body limbs reciprocally oscillate in a way not significantly affecting BCoM trajectory, and thus the external work was unable to account for the extra work done to move them. To include this component into the overall mechanical energy balance, the internal work ($W_{INT}$) was introduced [12,14]: by interpreting König's theorem of total kinetic energy for multi-segment systems, $W_{INT, K}$ was defined as the work needed to increase the kinetic and rotational kinetic energy of body segments, with their (linear) speed expressed as relative to BCoM. Such a 'kinetic' internal work, measured throughout three-dimensional motion capture of joint landmarks, was assumed to be a separate entity with respect to $W_{EXT}$. Since then, the total mechanical work ($W_{TOT} = W_{EXT} + W_{INT,K}$) [15,16] contributed to explain the metabolic cost of locomotion in several experimental conditions [1,4,8,17–20]. The scientific community has been critically debating that approach since the 1980s (e.g. [21,22]) but no

unbiased, alternative procedure has yet appeared to better assess the overall mechanical energy balance of limb-based locomotion. Here is not the place to illustrate the reasons why the original, 'kinetic' internal work is still the best approximation available for that problem. However, there is another important component of the internal work of locomotion, much less questionable from the theoretical point of view, which has never been considered so far. The 'frictional' internal work ($W_{INT,F}$), namely the work needed to overcome internal friction among moving tissues (joints, surrounding muscles and connective/fat tissue), remains undetectable by current biomechanical gait methodologies such as motion analysis/ground reaction forces/joint power-based calculation (as an example, a good actor can move upper limbs as to mimic the movement of two ideal reciprocal pendula—i.e. with no $W_{EXT} + W_{INT,K}$ apparently done—by properly using muscle activity, even when wearing a heavy overcoat that increases shoulder joint friction, resulting in an underestimated energy balance).

Amazingly enough, in the past, the work against 'internal friction' has been mentioned [12,14]. It was confined, however, to the friction inside muscle tissue; an effect already accounted, in a sense, by the muscle force/velocity relationship and reflected in the efficiency concept. In the successive decades, several papers appeared on isolated *ex vivo* body joints (e.g. [23–25]) where their very slightly damped rotation was measured. These results involuntarily supported/diffused the belief that joint friction could not remarkably affect the overall mechanical (and metabolic) energy balance of body movement and locomotion. Rather, everybody passively swinging a lower or upper limb realizes that, after a while, the oscillation comes to a stop. Such an energy dissipation, or negative work, issued by some 'compound' friction in a motion pattern that is, differently, continuously sustained during our gaits, has to be paralleled by some extra positive work done by muscles that should be assessed and incorporated in $W_{TOT}$. *In vivo* joint friction is 'compound' by nature, as it is not just a matter of articular surfaces, but also ligaments, capsules, tendons, bursae, muscles, fasciae and other tissues interacting nearby that contribute to further dampen limb oscillation. The relevance of this neglected component of the energy balance extends beyond the enhancement of understanding the basic physiology of healthy locomotion, with potential insights into impaired body movement up to joint prosthetic implants.

While those considerations were enough to initiate a study on limbs behaving like passively damped pendula, lower limb motion during stance configured a much more complex condition, which could not be ignored. The hip joint is exposed to a compressive stress, operated by the rest of the body weight (the trunk, head and the other three limbs, mainly) along an inverted pendulum trajectory during stance. Such a condition (1) is very different from passive swinging limbs (involving tensile stress within the joints), and (2) since synovial cartilage is exposed to high and variable compression forces, the damping factor could be higher than in swinging limbs and, due to the combined effects of potential deformation of that tissue and variable viscosity of synovial fluid (e.g. [25]), configure a deviation from the dry (Coulomb) to the viscous friction model and show load dependency. All these aspects of human joint friction, and their role in locomotion mechanics and energetics, have never been studied before.

Thus, aims of this study are (a) to measure the spontaneous decay of upper and lower limbs oscillations during passive swings as straight pendula, (b) to measure a (single) spontaneous oscillation of differently loaded, upside-down lower limb while pivoting on the hip joint as a passive inverted pendulum, (c) to estimate, throughout a custom-built mathematical model, the average damping coefficients in the three conditions, and (d) to provide an estimate of the 'frictional' internal work of locomotion based on those results.

## 2. Material and methods

### (a) Experimental protocol

Twelve males participated in this study (age $26 \pm 3$ years; stature $1.79 \pm 0.08$ m; body mass $73.0 \pm 6.7$ kg; mean ± s.d.). Each subject performed seven passive trials: straight unloaded oscillations of upper and lower limbs (five repetitions for each trial); and lower limb acting as an inverted unloaded and inverted loaded pendulum with loads of +4, +8, +10 and +12 kg distributed along the leg length (10 subjects; 10 repetitions for each trial). A total of 655 trials were analysed. Inverted limb oscillations were obtained by having subjects in an upside-down posture on a modified chair (figure 1) where shoulders were supported and the (loaded or unloaded) limb was impulsively pushed by an operator as to reach enough speed to cross the vertical axis; its motion was then stopped by a trajectory constraint positioned at the end of the investigated trajectory.

Oscillatory kinematics were recorded using seven infrared, three-dimensional motion cameras (Vicon Motion Systems, UK) sampling at 200 Hz. In both straight and inverted pendulum trials, subjects wore a brace in order to keep the limbs straight during the whole oscillation and were fitted with three reflected markers placed at the hip, knee and ankle joint centres. Additionally, in the loaded inverted trials, markers were placed at the centre of the added masses. Using the three-dimensional position of each marker in accordance with Dempster anthropometric tables [26], the proximal distance to the overall centre of mass and the radius of gyration of the investigated limb were calculated for each trial type. By using the parallel axis theorem, inertia parameters of loaded limbs were obtained. Phase planes of limb oscillation were visually inspected to ensure no muscular influence and, when active muscle contraction was apparent as a deviation from the smooth expected trajectory of straight or inverted damped pendulum (figure 2), the trial was discarded. Thereafter, the processed data were used to infer the damping within the joint according to the following theory.

### (b) The theory

The following (the detailed version is integrally reported in electronic supplementary materials) deals with the dynamics of a pendulum affected only by gravity and by a viscous damping torque: the first part will illustrate the mathematical solution for (small) passive oscillations of a straight pendulum starting from a manually imposed initial angle, while the second part will consider an inverted pendulum moving upward after a manually provided impulsive push (see Experimental protocol).

#### (i) Damped straight pendulum

A viscously damped ($b$, N m s rad$^{-1}$) pendulum of mass $m$ and length $R$ from the pivoting point, moves ($\ddot{\theta}$ is angle acceleration) according to the moment conservation law:

$$I\ddot{\theta} = \sum T \tag{2.1}$$

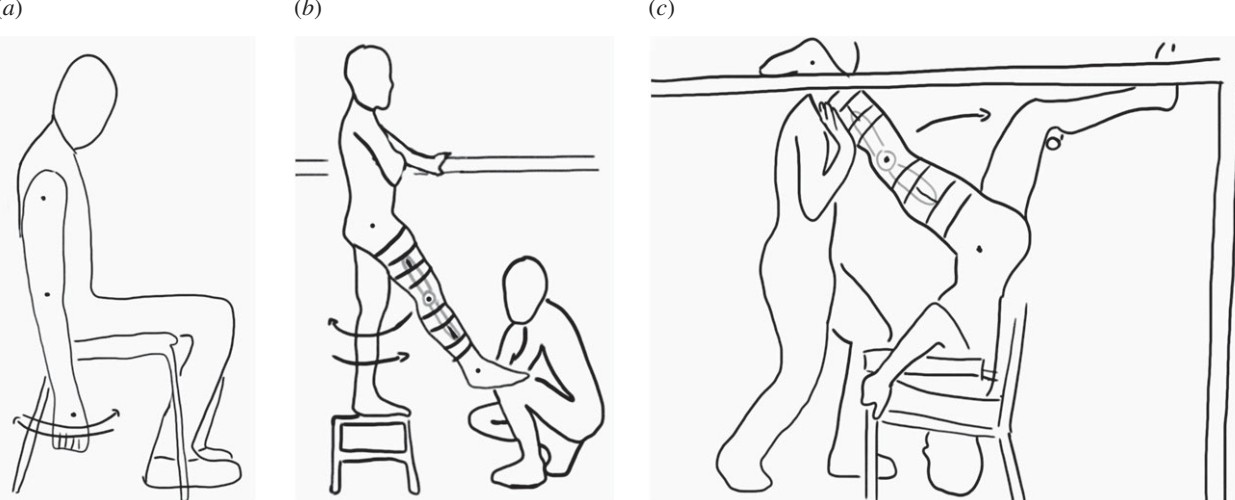

**Figure 1.** Experimental set-up for the unloaded straight pendulum of the upper (*a*) and lower (*b*) limbs. The unloaded/loaded inverted pendulum test (corresponding to the loaded stance phase in the gait cycle) is shown in (*c*). The knee brace (black and grey lines) prevents knee joint to flex. The built rack ensured safety by limiting the limb excursion angle throughout, reflective markers (black circles) are located on relevant body landmarks to ensure accurate capture of the limbs' movement.

and depends on the moment of inertia, and gravitational and damping torques.

By assuming $\sin \theta \approx \theta$ (for small oscillations), this translates into

$$\ddot{\theta} = -\frac{b}{m\,R_g^2}\,\dot{\theta} - g\,\frac{R}{R_g^2}\,\,\theta, \tag{2.2}$$

i.e. a second-order linear ordinary differential equation (ODE) where $Rg$ is radius of gyration about the pivot of the pendulum, $\theta$ and $\dot{\theta}$ are angle and angular speed, respectively.

The general solution of the second-order linear ODE (equation (2.2)) is

$$\theta(t) = C_1\,e^{r_1\,t} + C_2\,e^{r_2\,t}, \tag{2.3}$$

where $C_1$ and $C_2$ will be set according to the initial conditions of the pendulum in terms of $\theta_0$ (rad) and $\dot{\theta}_0$ (rad/s), while $r_1$ and $r_2$ are compound terms of its damping, gravity and physical/inertial characteristics. Depending on those last variables, the solution models overdamped, critically damped or underdamped oscillations, the last being the case for our upper or lower limb passively swinging (figure 3).

After extensive substitution and rearranging, pendulum angle time course corresponds to

$$\theta(t) = K\,e^{-(A/2)t}\,\sin(\omega\,t\,+\,\phi) \tag{2.4}$$

where amplitude ($K$), oscillation damping ($-A/2$), frequency ($\omega$) and phase ($\phi$) are represented by compound set of the quoted variables. In the extended form of equation (2.4), namely equation (23) in electronic supplementary materials, the damping factor $b$ is the only unknown variable, since initial conditions ($\theta_0$, $\dot{\theta}_0$) are experimentally set and all other symbols refer to anatomical characteristics and inertial properties of subjects' limbs (+ load for the inverted pendulum, see below).

The easiest procedure is to extract from experimental oscillations the straight pendulum angles and timing at repeated swing inversions, rectify them (see circles in figure 3) and perform an exponential ($y = p\,e^{qt}$) regression (the 'true' one; e.g. Labview, National Instruments, US, or Grapher, Apple Computers, US), not the linearized version (Excel, Microsoft, US; see *Statistical procedure*), providing the exponent coefficient $q$ ($= -A/2$ of equation (2.4)) from which $b$ can be obtained as

$$b = -2\,m\,R_g^2\,q. \tag{2.5}$$

### (ii) Damped inverted pendulum

It is worth bearing in mind that, differently from the straight pendulum, only a small portion of just one swing will be experimentally available for investigation of damping in (unloaded/loaded) lower limb behaving as an inverted pendulum. The above procedure for solving second-order linear ODE had to be adapted to the new condition and to benefit from the same simplification. Equation (2.1) is again the starting point, with small changes in the gravitational torque.

Here, introduced changes in equation (2.2) lead to a more manageable solution:

$$\theta(t) = C_1\,e^{M\,t} + C_2\,e^{N\,t}, \tag{2.6}$$

where $M$ and $N$ compound variables (of damping, gravity and physical/inertial characteristics) with values associated with an overdamped condition dynamics. Equation (2.6) and its differentiation allow to estimate the value of $C_1$ and $C_2$ according to the initial conditions $\theta_0$ and $\dot{\theta}_0$, leading to the closed-form solution equations.

In order to estimate damping factor $b$, here from a single swing, a different analysis strategy was developed, based also on the knowledge of the final conditions at the end of the experimental trajectory ($\theta_{end}$ and $\dot{\theta}_{end}$) and other derived variables, as the time duration of the oscillation from the initial angle to the final one ($t_{swing}$) and the timing of the minimum angular speed ($t_{\dot{\theta}_{MIN}}$). Further details on the analysis strategies are reported in electronic supplementary materials.

Finally, the data analysis leading to the relevant damping coefficients in this investigation can be summarized as follows. Straight Pendulum—after checking for passive oscillations the angular peak decay in successive swings was processed. Inverted pendulum—by knowing the initial and final pendular state and based on its inertial properties when unloaded or loaded pendulum it is possible to deduce the damping coefficient that reproduces the experimental oscillations or limb swings.

### (c) Statistical procedure

The accuracy and robustness of the illustrated data analyses for straight and inverse pendula have been checked by generating 'reference' angle time courses where inertial properties, initial values for the dynamics and the damping factor $b$ were imposed (Working Model, Knowledge Revolution, US). Then those data were fed into the two processing schemes to verify accuracy and precision of $b$ estimate.

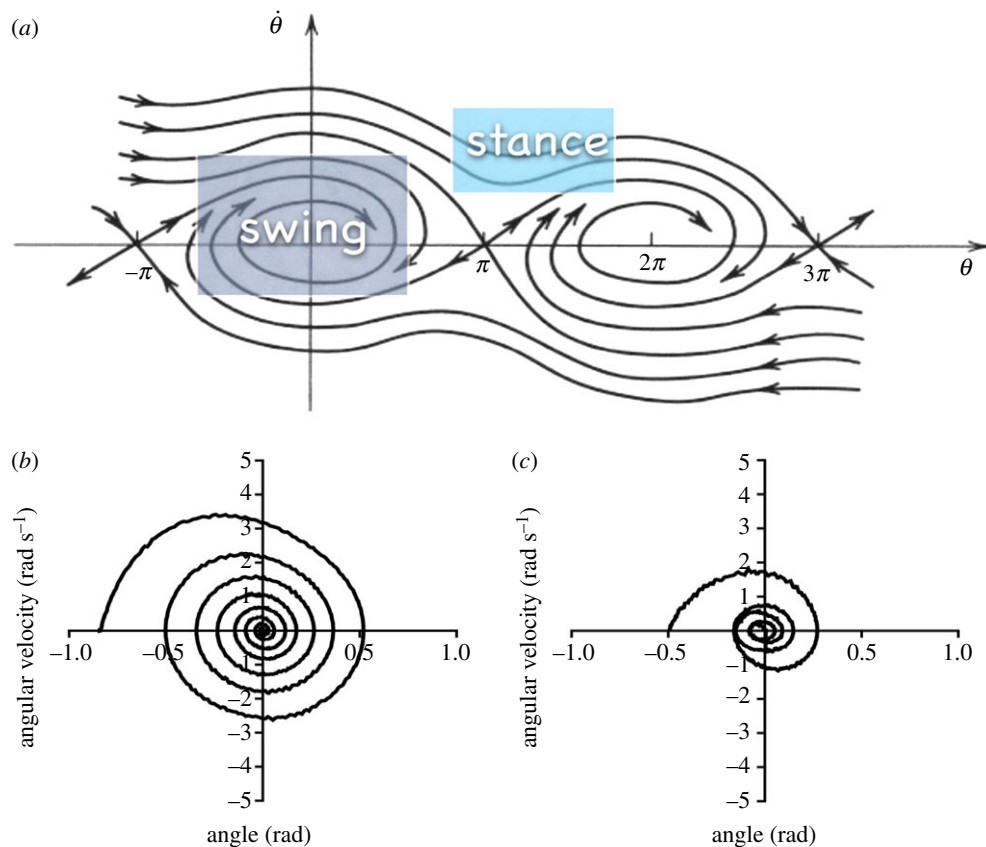

**Figure 2.** (a) A typical phase-plane (angle speed versus angle) of a damped pendulum, with each curve corresponding to different initial conditions ($\dot{\theta}_0$, $\theta_0$): independently from them, the (stable) pendulum destiny (mathematical attractor) is $\dot{\theta}_{end} = 0$ and $\theta_{end} \approx 2k\pi$, with $k$ integer, which could be also reached after complete rotations about the fulcrum. The area denoted as swing contains typical trajectories for a straight damped pendulum (corresponding in this paper to upper limbs in all locomotion phases and lower limbs during their swing phase only), while the one denoted as stance shows the behaviour of a damped inverted pendulum (corresponding here to lower limbs during their stance phase). Lower Panels. Experimental curves of naturally damped limb oscillations, straight pendulum position: (b) the data represented by this curve were accepted and processed since the spiral trajectory does not show any apparent deviation from the theoretical expectations (see a); (c) these data were discarded because the trajectory suggests the intervention of involuntary muscle contractions forcing not to follow the homogeneous spiral path. (Online version in colour.)

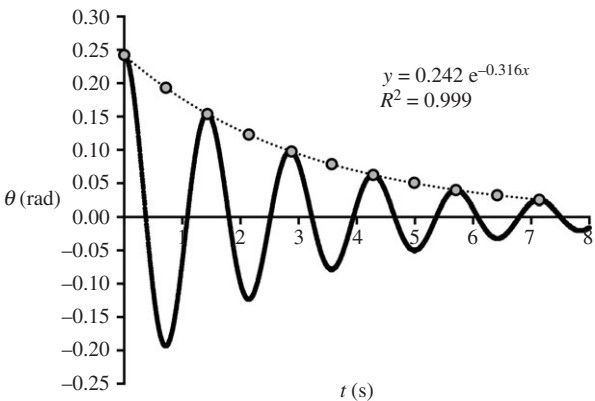

**Figure 3.** A typical oscillation damping of a straight pendulum with a viscous rotational friction located in the pivot. The damping coefficient is estimated by performing an exponential regression (see *Statistical procedure*) on peaks and (rectified) troughs (grey circles). Equations (24) in electronic supplementary materials and (2.5) describe how to obtain it from the resulting exponent coefficient (−0.316 in the illustration).

Exponential regression for analysing the oscillation peak angles in the straight pendulum experiments deserves some extra comments. Different software packages provide exponential regression analysis according to different optimization criteria, resulting in different estimates for the same data file processed. Although the general fitting algorithm is based on least-squares method (LSM), the packages differ on which distance has to be minimized. Microsoft Excel, by applying 'Add Trendline' after having selected a data series in a graph (and choosing Exponential in the setting window), minimizes the sum of squared distances between the logarithm of the vertical coordinate of experimental points and the fitting line (which, in a semi-logarithmic graph, corresponds to the exponential curve in the originally linear scale plot). In this way, the zone in the dataset where very small values appears (but that would happen also with very high values) will have a greater 'weight' in generating the finally fitting exponential, therefore often resulting in remarkable visual discrepancies between the dataset and the regression curve. Other software packages, as Labview (National Instruments, US, via a provided 'vi' (virtual instrument)) and Grapher (Apple Computers Inc., US, where users can custom design their own regression equations), apply LSM to the minimization of the sum of the vertical distances between experimental data and predictions on the 'final' regression exponential curve (in linear scale plot). This procedure 'equalizes' the importance of distances along the entire dataset range. In synthesis, the two procedures provide different predictions when applied to the same data. The decision about which to adopt has to be found in the need to minimize linear or logarithmic distances between points and their predictions. In this paper the former method has been used because the designed physical/mathematical model leads to an exponentially decaying time course of angle oscillation peaks in a linear scale plot where there is no issue in dealing with logarithmic distances.

The same data from simulated straight pendulum were fitted with the two LSM, and it was found that the sum of residuals was 97% lower by using the 'linear' method than with the 'linearized (via logarithmic transformation)' statistics of the exponential regression. Even visually, the 'unlinearized' method (Grapher, Labview) provided exponential curves much better interpolating the experimental data along the whole operative range; thus, as mentioned, it was adopted to analyse all straight pendulum measurements.

Phase planes, i.e. graphs showing the straight pendulum oscillation in terms of angular speed versus angle, were used to exclude trials where muscle activity was involved. In a passive damped pendulum phase planes show a spiral trajectory leading towards zero angle (with respect to the vertical) and zero angular speed (figure 2). Any apparent deviation from a regular spiral path was considered as produced by an involuntary muscle contraction.

The inverted pendulum analyses (Method 1: electronic supplementary material, equations (33) and (34); Method 2: electronic supplementary material, equation (35)) revealed to be more critical: even processing simulated data provided good predictions of b only when initial, final conditions and their timing were very accurately measured. Particularly, the timing had to include the half time step caused by angle differentiation (to obtain angular speed), and initial angle had to be the average of the first two data points. When these precautions were adopted, the b estimates fell within 2% of the imposed value for the inverted pendulum simulation, with a coefficient of variation of less than 1%.

## (d) Energy dissipated during a locomotory cycle, and the associated metabolic cost to overcome the damping of the limbs

The mechanical energy necessary to maintain a periodic oscillatory movement

$$\theta = A_h \, \sin{(\omega \, t)}, \tag{2.7}$$

where $A_h$ is half the angle range and $\omega$ is the frequency coefficient ($= 2 \, \pi \, f$, with f in Hz), of a damped pendulum with viscous friction b (J m s rad$^{-1}$) is calculated by integrating over one cycle the mechanical power equation of energy dissipation, obtained by multiplying angular speed (i.e. the derivative of equation (2.7)) by the damping moment (equal to the product between angular speed and damping coefficient).

Then, by estimating the distance travelled during one gait cycle as a function of the leg length, the mass-specific mechanical cost of transport to overcome internal friction of all the four limbs in the body ($C_{\text{mifa}}$, J kg$^{-1}$ m$^{-1}$) is given by:

$$C_{\text{mifa}} = \frac{\pi^2 \, B}{8 \, m \, R_L^2} \bar{v} \tag{2.8}$$

with $B = \sum_1^4 b_i$, where $b_i$ are the damping coefficients of each proximal joint of the limbs. Equation (2.8) assumes the four limbs of the same length ($R_L$) and spanning the same angle range (2 $A_h$) during locomotion.

Actually, to consider all the relevant energy dissipation of locomotion, the summation building up the variable B should include ankles (which were not investigated in this study). Schematically, these are the coefficients to be included:

$b_1$ = upper right limb, from unloaded swinging experiments, full cycle
$b_2$ = upper left limb, from unloaded swinging experiments, full cycle
$b_{3,1}$ = lower right limb, from unloaded swinging experiments, half cycle

$b_{3,2}$ = lower right limb, from loaded inverted swinging experiments, half cycle
$b_{4,1}$ = lower left limb, from unloaded swinging experiments, half cycle
$b_{4,2}$ = lower left limb, from loaded inverted swinging experiments, half cycle
$b_{5,1}$ = right ankle, from body inverted swinging experiments, half cycle
$b_{5,2}$ = left ankle, from body inverted swinging experiments, half cycle.

These eight coefficients can be reduced to four ($B = b_{\text{U}} + b_{\text{L,U}} + b_{\text{L,L}} + b_{\text{A}} = 2 \, b_1 + b_{3,1} + b_{4,1} + 0.25 \, b_5$) by considering that:

(a) $b_1 = b_2$, thus $b_{\text{U}} = 2$ upper limb, unloaded swinging experiments, full cycle
(b) $b_{3,1} = b_{4,1}$, thus $b_{\text{L,U}} =$ lower limb, unloaded swinging experiments, full cycle
(c) $b_{3,2} = b_{4,2}$, thus $b_{\text{L,L}} =$ lower limb, loaded inverted swinging experiments, full cycle
(d) $b_{5,1} = b_{5,2}$, thus $b_{\text{A}} =$ ankles, loaded inverted swinging, while experiments (will) reflect the effects of 2 ankles ($b_5$) while acting only during half a cycle, thus $b_{\text{A}} = 0.25 \, b_5$, full cycle.

## 3. Results

Experiments with limbs as straight pendula showed that the choice of a viscous damper in the model was justified: oscillation amplitude vanished exponentially (viscous friction) rather than linearly (dry friction), as shown in figure 4.

Damping coefficients for the straight pendulum experiments resulted the lowest for the upper limbs, revealing a friction of about 1/6 of the lower limb. Conversely, coefficients for the inverted lower limb were from about 118% to 300% higher than the latter, depending on the added load, as shown in figure 5.

The combined damping of all joints (needed for equation (2.8), no ankles included) is $B = 11.94$ N m s rad$^{-1}$, thus the mass-specific **m**echanical cost of transport related to **i**nternal **f**riction for **a**ll limbs ($C_{\text{mifa}}$, J kg$^{-1}$ m$^{-1}$) is

$$C_{\text{mifa}} = 0.242 \, \bar{v}. \tag{3.1}$$

Such a relationship (i.e. $W_{\text{INT,F}}$, in the units of the mechanical cost of transport, versus average progression speed) is shown in figure 6, together with previous experimental data [27] and model predicting equation of 'kinetic' internal work ($W_{\text{INT,K}}$, in the same units of the mechanical cost of transport) of walking and running at different speeds.

## 4. Discussion

The specific aim of this paper was to measure the frictional damping, *in vivo*, of the most relevant human joints during passive oscillations. The underlying aim was to complete the mechanical energy balance of locomotion by incorporating the energy dissipation involved in limb oscillation. This unprecedented attempt is based on the idea that since limbs are kept in continuous oscillation during the different gaits, whereas they tend to stop when moving passively, the energy dissipated by joint friction, undetectable by gait analysis methods, will have to be provided by muscles in excess from the role in propulsion. In the following, the discussion will separately focus on methodological aspects

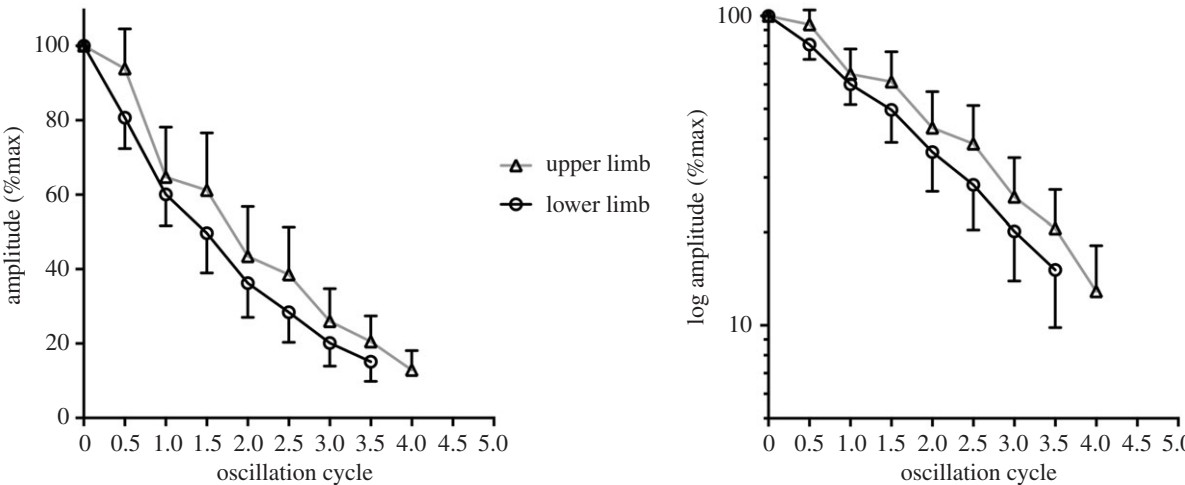

**Figure 4.** The decay of angle excursion, here represented as percentage of initial values, versus oscillation cycle number in straight unloaded swing of upper and lower limbs. The right panel checks for the viscous damping hypothesis: an exponential decay, shown in the left panel, becomes a linear decay when logarithm transformed peak angles are shown (data are mean ± s.d.).

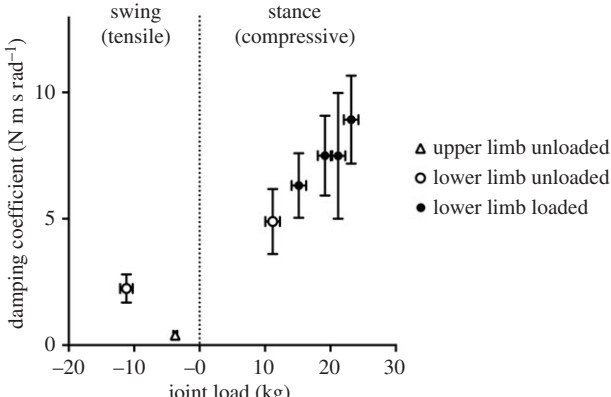

**Figure 5.** The damping coefficients measured from upper (triangles) and lower (circles) limbs during passive unloaded straight (empty symbols) and loaded inverted (solid symbols) pendulum oscillations are displayed as a function of the joint load. Load values refer to the mass (weight, actually) experienced by single joints in tensile (limb mass for straight pendula) and compressive stress (limb + load mass for inverted pendulum) (data are mean ± s.d.).

and on implications of the present results in locomotion mechanics/energetics.

## (a) Measurements

Although body limbs behave very much as simple pendula, modelling their passive motion depends on the friction type involved. The mathematical solution here developed is for viscous damping, as synovial fluid and its viscosity was initially thought to be the prevalent determinant of the overall friction. Actually, due to joints surrounded by many other tissues and to the upper arm/thorax and between thighs skin interaction, also dry friction or a mixed model could better explain the observed oscillations. However, the time course of peak angle decay can reveal whether the prevalent damping is viscous (exponential) or dry (linear) [28], and the experimental data seem better fit a curvilinear model (figure 4). In addition, other constraints such as asymmetrical anatomical range limits (e.g. shoulder endpoint during backward and forward swings) could alter the mechanical response and suggest to

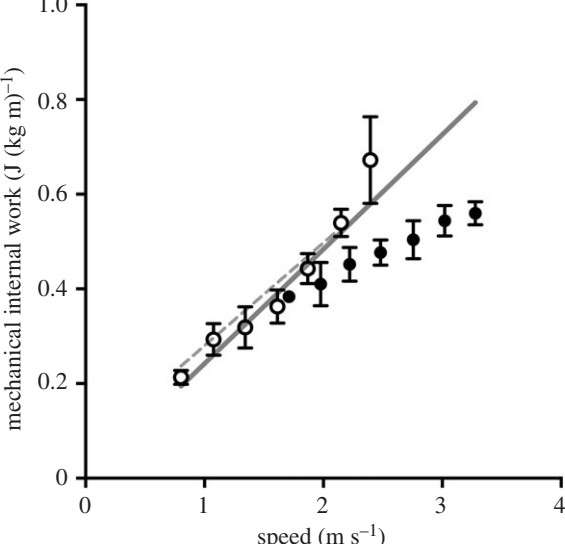

**Figure 6.** The mechanical internal work due to joint friction of all limbs ($C_{mifa}$, equation (3.1), in cost of transport units, corresponding to $W_{INT,F}$ (J) divided by body mass (kg) and by the distance travelled (m)) is represented by the continuous straight line. Experimental data of the other component of $W_{INT}$, namely $W_{INT,K}$, previously published [27] for walking and running (open and closed circles, respectively) as a function of locomotion speed are shown for comparison. The dashed line corresponds to equation (7) of a $W_{INT,K}$ model formulated in that paper.

separately analyse forward and backward swing phases. Even sometimes observing consistently (but slightly) lower peak angles for the backward swings only (figure 4), the two peak decay series resulted in very similar trends, thus with superimposable friction coefficients.

Passive swing experiments showed that upper limbs are exposed to much lower friction than lower ones; this is partly expected on the basis of their much lower muscle/tissue mass near to the proximal joint. Also, inverted lower limb experiments indicate that the higher the load (even at +0 kg the articular joint works in compression due to the leg mass) the greater the damping coefficients. This supports the hypothesis of a deformation of the articular cartilage under compressive load that could alter the damping effect during

the inverted swing of the leg (corresponding to the stance phase in locomotion). The complexity of the investigation protocol, particularly in the attempt to stiffen the knee joint, limited the extra load to a maximum of +12 kg (118 N), while the average *in vivo* load during walking is about +50 kg. Future studies will show whether the friction coefficients will increase further at higher loads or reach a plateau value similar to the maximum recorded in this study.

Such a potential underestimation of friction during (the stance phase, particularly, of) locomotion is partially mitigated by considering that all the limb oscillations were passive, thus reflecting friction caused also by structures, such as flexing/extending muscles, normally active in the different gait phases.

Another bias is potentially associated with how the cumulative damping ($B$) was obtained by summing the single joint coefficients. During experimental sessions care has been paid to have segments hanging just a little aside from the rest of the body, in order to 'see' the viscous damping effect of just internal and nearby tissues of the joint. This approach surely underestimated the (likely small) component of dry friction due to sliding of skin sheets between upper arm and the chest, and of inner skin surfaces between the proximal zones of the two thighs. It looks quite complex, at present, to assess skin friction alone, but it could be predicted that the incorporation of this component in the overall damping (of walking and running) be higher for lower limbs: force to overcome dry friction are proportional to the sliding area, which is greater, and mechanical power is proportional to sliding speed, which is double the one for upper limbs (relative to average progression speed, the two thighs move in opposed direction).

## (b) Implications for locomotion

As mentioned in the Introduction, mechanical external work is often the crucial component of total mechanical work but cannot explain *per se* the whole metabolic cost of locomotion. Despite all the theoretical criticisms, the inclusion of internal work (estimated throughout limb kinetics, $W_{INT,K}$) so far proved to establish a solid link between mechanics and energetics of human and animal movement in different conditions (for walking, see [1,4]).

A particularly interesting case is cycling, where the metabolic cost estimated is neither accurate nor precise when $W_{EXT}$ is the only predictor [29,30]. The inclusion of a pedalling frequency-related term, which is inherently related to the $W_{INT}$ concept [7,31,32], remarkably increases metabolic prediction reliability [19,30]. However, $W_{INT,K}$ has been demonstrated to be almost negligible in cycling due to the mechanically linked motion of the two lower limbs, and it was suggested that the crucial determinant of cycling economy, i.e. the metabolic equivalent of that total work, could be found in a different form, also pedalling rate related, of $W_{INT}$, namely the one due to friction internal to the body [30]. With the actual results, albeit lacking the inclusion of knee and ankle (and thigh/saddle interaction) damping coefficients, we are now in the position of confidently suggesting that $W_{INT,F}$ is the solution to the cycling energy balance problem. Also, the inclusion of the pedalling frequency as a crucial determinant in the prediction of metabolic cost of cycling, re-enforces the present rationale: a higher pedalling frequency implies higher limbs oscillation speed that, because of the prevalent viscous (speed-dependent) joint friction,

means a higher damping torque and, ultimately, higher mechanical work done and metabolic energy spent.

In terrestrial, limb-based locomotion similar considerations are not equally at reach. Walking and running (but also skipping) experimentally showed a consistent $W_{INT,K}$ (in a range of 30–60% of $W_{EXT}$ depending on speed; e.g. [4]) about which no investigation provided so far workable evidences for reducing it to a negligible component of the total mechanical work.

It is interesting to estimate the effects of joint friction during walking at about $1.4\,\mathrm{m\,s^{-1}}$, which is close to the metabolically optimal speed. Upper limbs show a small, almost negligible damping, resulting in 7% of the entire frictional internal work ($W_{INT,F} = 0.322\,\mathrm{J\,(kg\,m)^{-1}}$). Most of $W_{INT,F}$ (73%) is done during the stance phase, at the level of lower limb joints. Our preliminary, approximate model (equation (3.1)) indicates that the frictional internal work is a linear function of speed, quantitatively close to the 'kinetic' internal work ($W_{INT,K}$, [27]). The only previous investigation to compare part of our results is about actively swinging the lower (unloaded) limb back and forth at different frequencies while recording torque/angle and metabolic consumption necessary to maintain oscillation excursion [33]. The mechanical power obtained by those authors, despite using a different methodology, is of comparable order of magnitude of (just) the swinging lower limb component of the power necessary to counteract overall internal friction of the four limb joints in the present investigation.

These novel results challenge the mechanical (and metabolic) energy balance of limb-based locomotion, as it was known so far. Differently from cycling, it is very unlikely that a future revision of $W_{INT,K}$ of walking and running will make it vanish, due to a more solid theoretical basis sustaining that model approximation. The successive step, then, will be the attempt to understand a total (positive) mechanical work that is higher than previously reported because of an additional component ($W_{INT,F}$). Such an inflated $W_{TOT}$ and its effects in providing, for the same metabolic consumption, unrealistically high efficiency values could be possibly explained by reconsidering the influence of mechanical energy (storage and) release, which could be higher than thought both in bouncing gaits as running, but also in walking (evidences can be found in [34–36]). On the other hand, in studies where the metabolic equivalent of BCoM deceleration and lowering (i.e. negative $W_{EXT}$) and of segment decelerations (i.e. negative $W_{INT,K}$) need to be included in the (almost fully comprehensive) total energy balance of locomotion [1,2,5], the present findings will contribute to decrease the measured negative work by the expected amount due to joint friction. As it is known that negative work efficiency of muscle contraction is about five times higher than the one for positive work [37], though, this estimate refinement is supposed to be minor.

## 5. Conclusion

Main joints friction during locomotion-related limb oscillations devoted to simulate swing and stance phases has been evaluated *in vivo*. A mathematical model of damped pendulum dynamics and data analysis revealed that a remarkable fraction of the total mechanical work of

locomotion has to be done to overcome friction at pivoting points of the limbs. Such a frictional internal mechanical work complements the internal work needed to accelerate body segment with respect to the BCoM depending on the locomotion type: in (stationary) cycling it is actually the only form of internal work that, together with the external one, is capable to explain the relationship between metabolic expenditure and pedalling frequency. During walking or running, both forms of mechanical internal work (the frictional being more reliable) cumulate with the external work to reach a total amount that potentially moves efficiency far above the reference value for muscles, encouraging to revise the contribution of elastic energy saving via tendons and other structures even in non-bouncing gaits. This is an issue future research will need to address.

The presented theoretical and experimental methodology allows to design a cascade of studies extending, complementing and deepening the scope of this investigative framework: (a) joint friction differences between genders and across the lifespan, (b) knee and ankle inclusion in the list of measured joints, (c) inverted pendulum (hip simulation during stance) with loads better approaching the real locomotor condition, (d) the severely obese patient as the upmost limit of joint friction and its effects on locomotion, and (e) overall friction after articular prosthetic surgery to extrapolate the joint friction due to structures nearby the implant only, a novel and useful information for locomotion physiology.

Ethics. All procedures were approved by the Ethical Committee and subjects signed an informed consent, where all procedure and risks were fully explained.

Data accessibility. Data available from the UNIMI Dataverse: https://dataverse.unimi.it/privateurl.xhtml?token=d458c523-7665-437f-923e-086fac9d2930.

Authors' contributions. A.E.M. conceived and designed the research. A.P.M. and G.P. performed the experiments. A.E.M., A.P.M. and G.P. analysed the data. All authors contributed to interpretation of the results. A.E.M., A.P.M. and G.P. prepared the figures. A.E.M. drafted manuscript. All authors edited and revised manuscript. All authors have read and approved the final version of this manuscript and agree to be accountable for all aspects of the work in ensuring that questions related to the accuracy or integrity of any part of the work are appropriately investigated and resolved. All persons designated as authors qualify for authorship, and all those who qualify for authorship are listed.

Competing interests. We declare we have no competing interests.

Funding. This work was supported by 'Tariffario Laboratorio Analisi' Research Fund, 'Faculty of Medicine' of the University of Milan, n. 09-minetti 27747.

Acknowledgements. The authors appreciate the time and effort put in by all subjects.

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
