## [Reviewer comments · Proceedings of the Royal Society B: Biological Sciences]

Review History

RSPB-2020-0348.R0 (Original submission)

Review form: Reviewer 1 (Arthur D. Kuo)

Recommendation

Accept with minor revision (please list in comments)

Scientific importance: Is the manuscript an original and important contribution to its field?

Good

General interest: Is the paper of sufficient general interest?

Good

Quality of the paper: Is the overall quality of the paper suitable?

Good

Is the length of the paper justified?

No

Should the paper be seen by a specialist statistical reviewer?

No

Do you have any concerns about statistical analyses in this paper? If so, please specify them explicitly in your report.

No

It is a condition of publication that authors make their supporting data, code and materials available - either as supplementary material or hosted in an external repository. Please rate, if applicable, the supporting data on the following criteria.

Is it accessible?

N/A

Is it clear?

N/A

Is it adequate?

N/A

Do you have any ethical concerns with this paper?

No

Comments to the Author

This manuscript proposes that friction at the leg joints contributes substantially to the overall energetic cost of human locomotion. The experiment involves estimating friction from the leg acting as a damped pendulum, in both loaded and unloaded conditions. This is an interesting study, described in a clear manner. Few previous studies have examined the role of friction, and this manuscript provides good evidence to indicate that the role may be substantial. I believe this is well worth publishing, and provide a few basic suggestions for improvement below.

Introduction

Although the authors argue (p. 4) that frictional internal work "has never been considered so far," there is a key paper that argues precisely that friction should be important: Radhakrishnan (1998) "Locomotion: Dealing with friction" PNAS 95: 5448-5455. This is worth citing, does not take away from the present study's experimental contributions, and fact also supports the theoretical basis.

Others have also experimentally considered the role of friction in some animals, e.g.

https://www.researchgate.net/publication/255649895_Damping_And_Size_Insights_And_Biological_Inspiration.

The actual energetic cost of moving the leg as a pendulum has also been measured:

<https://jeb.biologists.org/content/208/3/439>. The present manuscript could provide additional insight on contributions to these previous measurements.

Materials and Methods

I commend the authors on the clear exposition of experiment and analysis. A small suggestion: The mathematical details of the damped pendulum take up a great deal of space (almost six pages of manuscript document), and are both quite complex for the non-mathematical reader, and too simple for those more experienced with pendulum dynamics. It is perhaps worth summarizing the concepts here, and relegating the details to an Appendix (or citing Wikipedia and other sources that cover much of the same material). This is not to take away from the careful work done here, nor the need to define the parameters used by the authors, but to enable the reader to understand the study with less distraction.

Discussion

Line 475 "lower limbs, differently from the upper ones, are mutually touching."

I was confused by this statement. Are the authors referring to the fact that lower legs both about a

single pelvis but upper limbs do not? I do not understand how articulation about pelvis would make a difference. Or does the statement mean that the legs brush or contact each other during locomotion? For many people, there is only incidental contact. Clarification would be helpful.

As cited above, the JEB article reports energetic cost of moving the leg back and forth, and current results could potentially lend insight and be compared with that cost.

Review form: Reviewer 2

Recommendation

Major revision is needed (please make suggestions in comments)

Scientific importance: Is the manuscript an original and important contribution to its field?

Good

General interest: Is the paper of sufficient general interest?

Acceptable

Quality of the paper: Is the overall quality of the paper suitable?

Good

Is the length of the paper justified?

No

Should the paper be seen by a specialist statistical reviewer?

No

Do you have any concerns about statistical analyses in this paper? If so, please specify them explicitly in your report.

No

It is a condition of publication that authors make their supporting data, code and materials available - either as supplementary material or hosted in an external repository. Please rate, if applicable, the supporting data on the following criteria.

Is it accessible?

Yes

Is it clear?

Yes

Is it adequate?

Yes

Do you have any ethical concerns with this paper?

No

Comments to the Author

I have considered the paper by Minetti et al. on 'Frictional internal work of damped limb oscillations'. Intrinsically, this is a very interesting paper. It provides the starting point for further research on an overlooked component of the mechanical work that must be delivered by the muscles during legged locomotion: i.e. the work needed to overcome internal friction. Based on simple but relevant experiments in which straight arms and legs are recorded while swinging

freely (hanging or as a 'loaded' inverted pendulum) the (velocity dependent) damping coefficient in play to explain the observed damped cycles (or part of a cycle in case of the inverted pendulum) were estimated (for several boundary conditions, loadings). This damping represents energy dissipation that, in cyclic locomotion, must be replenished cycle by cycle by the muscle and the cost of this in locomotion (in Joules/m/kg body mass) could be estimated from the damping coefficients. It turns out that this cost is comparable in magnitude to this related to the so-called kinetic internal work (required to move segments with respect the body centre of mass). The latter is known to amount 30 to 60% of the so-called external work (i.e. the work needed from the muscle to move the centre of mass itself). As such, overcoming internal friction seems to constitute an important component of the total mechanical (hence also metabolic) costs.

My major remark on this contribution concerns the heavy load of mathematics. I'm really lost in this part of the manuscript. Moreover, I wonder what the real use of the models is because the authors seem to rely in the end (if I understood correctly) on relatively simple regression methods to extract the damping coefficients from the experimental data on which their conclusions are based. If interpreted correctly, the models are used to construct a kind of theoretical 'behavioural' landscape depending varying boundary conditions and damping coefficients, after which it could be established where the experiments map (how? optimization modelling?) on that landscape (but the question remains why then return to the regressions?). Most probably, this remark originates from my too limited knowledge of, and skills in, maths. Since I cannot believe the maths are there just to overwhelm the maths-layman, (but interested biologist, physiologist, etc.), the line of thought should be much better explained (preferable in a more colloquial, verbal way, moving the equation deduction to an appendix or to supplementary material). On the other hand, if my initial impression, that the extensive presentation of the mathematical models is, to a large extent, redundant for the main message, is correct, why present these anyway?

Decision letter (RSPB-2020-0348.R0)

01-May-2020

Dear Professor Minetti:

I am writing to inform you that your manuscript RSPB-2020-0348 entitled "FRICTIONAL INTERNAL WORK OF DAMPED LIMBS OSCILLATION IN HUMAN LOCOMOTION" has, in its current form, been rejected for publication in Proceedings B.

This action has been taken on the advice of referees, who have recommended that substantial revisions are necessary. With this in mind we would be happy to consider a resubmission, provided the comments of the referees are fully addressed. However please note that this is not a provisional acceptance.

- 1) A 'response to referees' document including details of how you have responded to the comments, and the adjustments you have made.

- 2) A clean copy of the manuscript and one with 'tracked changes' indicating your 'response to referees' comments document.
- 3) Line numbers in your main document.

Sincerely,

Dr John Hutchinson, Editor
 mailto: proceedingsb@royalsociety.org

Associate Editor

Board Member: 1

Comments to Author:

Thank you for submitting your manuscript to Proceedings B. Both reviewers consider the manuscript to be an interesting and important addition to the field. However, both reviewers also query the heavy mathematical content of the main manuscript and whether this is suitable for the broad readership of PRSB, and I am inclined to agree with them. In addition to addressing their other comments, please consider moving much of the mathematical proof to the supplementary material. And, as suggested by Reviewer 2, consider replacing this section with an easier-to-follow verbal description of the model, that may be of more interest to our readership of organismal anatomists and physiologists.

Reviewer(s)' Comments to Author:

Referee: 1

Comments to the Author(s)

This manuscript proposes that friction at the leg joints contributes substantially to the overall energetic cost of human locomotion. The experiment involves estimating friction from the leg acting as a damped pendulum, in both loaded and unloaded conditions. This is an interesting study, described in a clear manner. Few previous studies have examined the role of friction, and this manuscript provides good evidence to indicate that the role may be substantial. I believe this is well worth publishing, and provide a few basic suggestions for improvement below.

Introduction

Although the authors argue (p. 4) that frictional internal work "has never been considered so far," there is a key paper that argues precisely that friction should be important: Radhakrishnan (1998) "Locomotion: Dealing with friction" PNAS 95: 5448-5455. This is worth citing, does not take away from the present study's experimental contributions, and fact also supports the theoretical basis.

Others have also experimentally considered the role of friction in some animals, e.g.

https://www.researchgate.net/publication/255649895_Damping_And_Size_Insights_And_Biological_Inspiration.

The actual energetic cost of moving the leg as a pendulum has also been measured:

<https://jeb.biologists.org/content/208/3/439>. The present manuscript could provide additional insight on contributions to these previous measurements.

Materials and Methods

I commend the authors on the clear exposition of experiment and analysis. A small suggestion: The mathematical details of the damped pendulum take up a great deal of space (almost six

pages of manuscript document), and are both quite complex for the non-mathematical reader, and too simple for those more experienced with pendulum dynamics. It is perhaps worth summarizing the concepts here, and relegating the details to an Appendix (or citing Wikipedia and other sources that cover much of the same material). This is not to take away from the careful work done here, nor the need to define the parameters used by the authors, but to enable the reader to understand the study with less distraction.

Discussion

Line 475 “lower limbs, differently from the upper ones, are mutually touching.”

I was confused by this statement. Are the authors referring to the fact that lower legs both about a single pelvis but upper limbs do not? I do not understand how articulation about pelvis would make a difference. Or does the statement mean that the legs brush or contact each other during locomotion? For many people, there is only incidental contact. Clarification would be helpful.

As cited above, the JEB article reports energetic cost of moving the leg back and forth, and current results could potentially lend insight and be compared with that cost.

Referee: 2

Comments to the Author(s)

I have considered the paper by Minetti et al. on ‘Frictional internal work of damped limb oscillations’. Intrinsically, this is a very interesting paper. It provides the starting point for further research on an overlooked component of the mechanical work that must be delivered by the muscles during legged locomotion: i.e. the work needed to overcome internal friction. Based on simple but relevant experiments in which straight arms and legs are recorded while swinging freely (hanging or as a ‘loaded’ inverted pendulum) the (velocity dependent) damping coefficient in play to explain the observed damped cycles (or part of a cycle in case of the inverted pendulum) were estimated (for several boundary conditions, loadings). This damping represents energy dissipation that, in cyclic locomotion, must be replenished cycle by cycle by the muscle and the cost of this in locomotion (in Joules/m/kg body mass) could be estimated from the damping coefficients. It turns out that this cost is comparable in magnitude to this related to the so-called kinetic internal work (required to move segments with respect the body centre of mass). The latter is known to amount 30 to 60% of the so-called external work (i.e. the work needed from the muscle to move the centre of mass itself). As such, overcoming internal friction seems to constitute an important component of the total mechanical (hence also metabolic) costs.

My major remark on this contribution concerns the heavy load of mathematics. I’m really lost in this part of the manuscript. Moreover, I wonder what the real use of the models is because the authors seem to rely in the end (if I understood correctly) on relatively simple regression methods to extract the damping coefficients from the experimental data on which their conclusions are based. If interpreted correctly, the models are used to construct a kind of theoretical ‘behavioural’ landscape depending varying boundary conditions and damping coefficients, after which it could be established where the experiments map (how? optimization modelling?) on that landscape (but the question remains why then return to the regressions?). Most probably, this remark originates from my too limited knowledge of, and skills in, maths. Since I cannot believe the maths are there just to overwhelm the maths-layman, (but interested biologist, physiologist, etc.), the line of thought should be much better explained (preferable in a more colloquial, verbal way, moving the equation deduction to an appendix or to supplementary material). On the other hand, if my initial impression, that the extensive presentation of the mathematical models is, to a large extent, redundant for the main message, is correct, why present these anyway?

Author's Response to Decision Letter for (RSPB-2020-0348.R0)

See Appendix A.

RSPB-2020-1410.R0

Review form: Reviewer 2

Recommendation

Accept with minor revision (please list in comments)

Scientific importance: Is the manuscript an original and important contribution to its field?

Excellent

General interest: Is the paper of sufficient general interest?

Good

Quality of the paper: Is the overall quality of the paper suitable?

Good

Is the length of the paper justified?

No

Should the paper be seen by a specialist statistical reviewer?

No

Do you have any concerns about statistical analyses in this paper? If so, please specify them explicitly in your report.

No

It is a condition of publication that authors make their supporting data, code and materials available - either as supplementary material or hosted in an external repository. Please rate, if applicable, the supporting data on the following criteria.

Is it accessible?

Yes

Is it clear?

Yes

Is it adequate?

Yes

Do you have any ethical concerns with this paper?

No

Comments to the Author

Dear Editor,

I have considered the revision of the paper by Minetti et al. on 'Frictional internal work of damped limb oscillations'. I'm still convinced about the importance of the results presented in this study, which is very well framed in the introduction. The presented findings may be relevant for proximate physiological, functional morphological, kinesiological and biomechanical

studies but, potentially also, for instance, for the proper understanding of the evolution of locomotor behaviour. Therefore, to attract as many readers as possible (and to convince them immediately about the main message: 'frictional work cannot be neglected'), I would personally still prefer to keep the body of the paper abstract of maths as much as possible. This does not mean I doubt its relevance, but I'm convinced that it will be beneficial for the readability of the paper if most maths are moved to an appendix or supplementary material. I don't see, for instance, what the benefit is of having equations 23 or 33-35 in the text. It seems more important to me to mention clearly what is aimed at with the experiments and models (something like "Knowing the initial pendular state (angular position and speed) and based on the morphometrics of the (straight and inverted) pendulum (inertial properties) it is possible to deduce the damping coefficient that reproduce the experimental oscillations or limb swings (for the mathematical details, see appendix or supplementary material) in order to maximize focus on the important results. Similarly, while definitely relevant, the discourse on what the appropriate regression procedure is (L205-230) is, according to me redundant for the body of the paper (but again, relevant for an appendix). This is, of course a personal view and preference. In its present format, I would also suggest to move the description of the experimental protocol to the beginning of Materials and Methods. The reason for this is that much of the theoretical explanations become more meaningful when the reader has already an idea of the experiments performed. This is especially true for what is written in L178-185. It's difficult to understand what is meant if one has no idea about what the 'inverted' experiments involve. There is also one thing that puzzles me. It concerns what is mentioned in L185-190. I'm confused about what is written here. I interpret this as if multiple damping coefficients can be selected by the algorithm that can all provide a proper representation of individual experimental results. According to me, there can be only one beta that results in the correct angular position and speed after time t-swing.

Decision letter (RSPB-2020-1410.R0)

26-Jun-2020

Dear Professor Minetti:

Your manuscript has now been peer reviewed and the reviews have been assessed by an Associate Editor. The reviewers' comments (not including confidential comments to the Editor) and the comments from the Associate Editor are included at the end of this email for your reference. As you will see, the reviewers and the Editors have raised some concerns with your manuscript and we would like to invite you to revise your manuscript to address them. The reliance on equations must be reduced, to better suit the audience of the journal.

When submitting your revision please upload a file under "Response to Referees" in the "File Upload" section. This should document, point by point, how you have responded to the reviewers' and Editors' comments, and the adjustments you have made to the manuscript. We

require a copy of the manuscript with revisions made since the previous version marked as 'tracked changes' to be included in the 'response to referees' document.

Research ethics:

Use of animals and field studies:

Please submit a copy of your revised paper within three weeks. If we do not hear from you within this time your manuscript will be rejected. If you are unable to meet this deadline please let us know as soon as possible, as we may be able to grant a short extension.

Best wishes,
Dr John Hutchinson, Editor
mailto: proceedingsb@royalsociety.org

Associate Editor Board Member

Comments to Author:

Thank you for submitting your revised manuscript to Proceedings B. We appreciate the effort that authors have put into revising their paper, and we note that the number of equations within the main body of text has now dropped significantly. The referee has only a single comment/question regarding the actual content of the manuscript. Otherwise, their outstanding concern still remains the 'readability' of the manuscript. Whilst we recognize this has already improved considerably, please take note of their suggestions with regard to the structure and narrative of the methods.

Reviewer(s)' Comments to Author:

Referee: 2

Comments to the Author(s).

Dear Editor,

I have considered the revision of the paper by Minetti et al. on 'Frictional internal work of damped limb oscillations'. I'm still convinced about the importance of the results presented in this study, which is very well framed in the introduction. The presented findings may be relevant for proximate physiological, functional morphological, kinesiological and biomechanical studies but, potentially also, for instance, for the proper understanding of the evolution of locomotor behaviour. Therefore, to attract as many readers as possible (and to convince them immediately about the main message: 'frictional work cannot be neglected'), I would personally still prefer to keep the body of the paper abstract of maths as much as possible. This does not mean I doubt its relevance, but I'm convinced that it will be beneficial for the readability of the paper if most maths are moved to an appendix or supplementary material. I don't see, for instance, what the benefit is of having equations 23 or 33-35 in the text. It seems more important to me to mention clearly what is aimed at with the experiments and models (something like "Knowing the initial pendular state (angular position and speed) and based on the morphometrics of the (straight and inverted) pendulum (inertial properties) it is possible to deduce the damping coefficient that reproduce the experimental oscillations or limb swings (for the mathematical details, see appendix or supplementary material) in order to maximize focus on the important results. Similarly, while definitely relevant, the discourse on what the appropriate regression procedure is (L205-230) is, according to me redundant for the body of the paper (but again, relevant for an appendix). This is, of course a personal view and preference.

In its present format, I would also suggest to move the description of the experimental protocol to the beginning of Materials and Methods. The reason for this is that much of the theoretical explanations become more meaningful when the reader has already an idea of the experiments performed. This is especially true for what is written in L178-185. It's difficult to understand what is meant if one has no idea about what the 'inverted' experiments involve. There is also one thing that puzzles me. It concerns what is mentioned in L185-190. I'm confused about what is written here. I interpret this as if multiple damping coefficients can be selected by the algorithm that can all provide a proper representation of individual experimental results. According to me,

there can be only one beta that results in the correct angular position and speed after time t -swing.

Author's Response to Decision Letter for (RSPB-2020-1410.R0)

See Appendix B.

Decision letter (RSPB-2020-1410.R1)

07-Jul-2020

Dear Professor Minetti

I am pleased to inform you that your manuscript entitled "FRICTIONAL INTERNAL WORK OF DAMPED LIMBS OSCILLATION IN HUMAN LOCOMOTION" has been accepted for publication in Proceedings B. Congratulations!!

Open Access

Paper charges

Sincerely,
Dr John Hutchinson
Editor, Proceedings B
mailto: proceedingsb@royalsociety.org

Appendix A

Reply to reviewers (*in Italic*)

Associate Editor

Board Member: 1

Comments to Author:

Thank you for submitting your manuscript to Proceedings B. Both reviewers consider the manuscript to be an interesting and important addition to the field. However, both reviewers also query the heavy mathematical content of the main manuscript and whether this is suitable for the broad readership of PRSB, and I am inclined to agree with them. In addition to addressing their other comments, please consider moving much of the mathematical proof to the supplementary material. And, as suggested by Reviewer 2, consider replacing this section with an easier-to-follow verbal description of the model, that may be of more interest to our readership of organismal anatomists and physiologists.

The mathematical proof and other very technical details of the Materials and Methods have been integrally moved to the new Supplementary Material zone. In the original section we now describe the whole mathematical process in 13 equations (out of 47) and the text has been edited to smoothly move through them. Equations in the main text maintain the original numbering as to provide a reference to their position in the integral sequence now in Supplementary Material. The authors express the wish to keep the Supplementary Material positioned at the end of the paper, rather than making it only downloadable as a separate file in the journal repository. This solution would simultaneously make the main paper more readable, as requested, and grant readers the access to the entire methodological process available in the same file.

Reviewer(s)' Comments to Author:

Referee: 1

Comments to the Author(s)

This manuscript proposes that friction at the leg joints contributes substantially to the overall energetic cost of human locomotion. The experiment involves estimating friction from the leg acting as a damped pendulum, in both loaded and unloaded conditions. This is an interesting study, described in a clear manner. Few previous studies have examined the role of friction, and this manuscript provides good evidence to indicate that the role may be substantial. I believe this is well worth publishing, and provide a few basic suggestions for improvement below.

Introduction

Although the authors argue (p. 4) that frictional internal work “has never been considered so far,” there is a key paper that argues precisely that friction should be important: Radhakrishnan (1998) “Locomotion: Dealing with friction” PNAS 95: 5448-5455. This is worth citing, does not take away from the present study’s experimental contributions, and fact also supports the theoretical basis.

The suggested paper is a very nice and extensive review of all sort of friction acting on the external part of animal and human bodies. It seems that no quote is made to the frictional effects of internal biologic structures to damp locomotion (and make it more expensive). Thus we now include that reference to say that most of the attention to friction has been directed to the external one.

Others have also experimentally considered the role of friction in some animals, e.g. <https://www.researchgate.net/publication/255649895> Damping And Size Insights And Biological Inspiration.

It is certainly a nice suggestion, as it tries to address comparative scaling of limb oscillation damping. Unfortunately, the mathematical model was not paralleled by measurements in animals of very different size (just cockroaches) to check the general message (that damping is relatively higher at smaller body size). It is also a pity that these conference proceedings paper were not followed by the promised and more detailed publication later on.

The actual energetic cost of moving the leg as a pendulum has also been measured: <https://jeb.biologists.org/content/208/3/439>. The present manuscript could provide additional insight on contributions to these previous measurements.

The suggested paper allows a partial comparison with our data and model. Doke et al. 2005 estimated and measured the mechanical work of freely and actively swinging the lower limb back and forth at different oscillation frequency. From their fig. 3B the measured mechanical work rate associated to a frequency of 0.9 Hz (typical for a walking speed of about 1.3 m/s) was about 0.125 W/kg. The steady state metabolic power for such a repeated leg swinging (fig. 4A) was about 1 W/kg, leading us to estimate an efficiency of 0.125. The present paper, namely from eq. 47 and fig. 7, predicts that the internal mechanical work to overcome friction for all the 4 limbs (2 swinging upper limbs, 2 half swinging lower limbs and 2 loaded half stancing lower limbs) at a walking speed of 1.3 m/s is 0.3 J/(kg m), i.e. a mechanical power of 0.4 W/kg. By considering that the summation of damping factors of the whole body sums up to about 12 N M s rad⁻¹ and the Lower Limbs Unloaded accounts for 1/4 - 1/5 of that value, the almost 1/3 estimate of mechanical internal work rate by Doke et al. with respect to our is of comparable order of magnitude. The discrepancy seems negligible when considering that the two papers follow completely different routes to achieve similar goals. A summary of this comment has been now incorporated into the Discussion.

Materials and Methods

I commend the authors on the clear exposition of experiment and analysis. A small suggestion: The mathematical details of the damped pendulum take up a great deal of space (almost six pages of manuscript document), and are both quite complex for the non-mathematical reader, and too simple for those more experienced with pendulum dynamics. It is perhaps worth summarizing the concepts here, and relegating the details to an Appendix (or citing Wikipedia and other sources that cover much of the same material). This is not to take away from the careful work done here, nor the need to define the parameters used by the authors, but to enable the reader to understand the study with less distraction.

We searched the Internet at the beginning of the study and we found that a close solution for a viscously damped straight and inverse pendulum with no massless rod was not found. That is why a complete solution of the ODE in the two conditions, together with the statistical/computational methodology to extract damping coefficients from experimental data was designed from scratch. We agree that the topic is quite basic for mechanical engineers.

By also coping with the concerns of both Reviewer 2 and Editor, we opted to summarize the mathematical process in the main text, and reduced the number of equations from 47 to 13 (while

maintaining the original equation numbering to allow tracking) and moved the integral process to Supplementary Material.

Discussion

Line 475 “lower limbs, differently from the upper ones, are mutually touching.”

I was confused by this statement. Are the authors referring to the fact that lower legs both about a single pelvis but upper limbs do not? I do not understand how articulation about pelvis would make a difference. Or does the statement mean that the legs brush or contact each other during locomotion? For many people, there is only incidental contact. Clarification would be helpful.

We agree that the paragraph could more intelligible. We meant that, in addition to the viscous friction inside and nearby the joint, some dry friction could occur between skin sheets of upper arm and the chest, and between internal surfaces of the proximal zones of the two thighs. During the experiments we took care of having the investigated segment hanging just a little aside from the rest of the body, in order to ‘see’ the damping effect of just internal and nearby tissues of the joint. For this reason, results could suffer from a small underestimation. An additional consideration which could add further bias is that those skin sheets slide on each other at a speed proportional to walking speed for upper limbs (by assuming that the trunk is travelling at constant velocity) proportional to twice that value for the lower limbs (as one segment is moving forward and the other backward). The difference between the two condition does not affect the dry (Coulomb) friction values (while we expect it to be greater in lower limbs due to the greater skin surface involved), but mechanical power to contrast such a damping component would be higher where sliding speed is greater. Part of this comment has been now incorporated into the Discussion.

As cited above, the JEB article reports energetic cost of moving the leg back and forth, and current results could potentially lend insight and be compared with that cost.

Done, see above.

Referee: 2

Comments to the Author(s)

I have considered the paper by Minetti et al. on ‘Frictional internal work of damped limb oscillations’. Intrinsically, this is a very interesting paper. It provides the starting point for further research on an overlooked component of the mechanical work that must be delivered by the muscles during legged locomotion: i.e. the work needed to overcome internal friction. Based on simple but relevant experiments in which straight arms and legs are recorded while swinging freely (hanging or as a ‘loaded’ inverted pendulum) the (velocity dependent) damping coefficient in play to explain the observed damped cycles (or part of a cycle in case of the inverted pendulum) were estimated (for several boundary conditions, loadings). This damping represents energy dissipation that, in cyclic locomotion, must be replenished cycle by cycle by the muscle and the cost of this in locomotion (in Joules/m/kg body mass) could be estimated from the damping coefficients. It turns out that this cost is comparable in magnitude to this related to the so-called kinetic internal work (required to move segments with respect the body centre of mass). The latter is known to amount 30 to 60% of the so-called external work (i.e. the work needed from the

muscle to move the centre of mass itself). As such, overcoming internal friction seems to constitute an important component of the total mechanical (hence also metabolic) costs.

My major remark on this contribution concerns the heavy load of mathematics. I'm really lost in this part of the manuscript. Moreover, I wonder what the real use of the models is because the authors seem to rely in the end (if I understood correctly) on relatively simple regression methods to extract the damping coefficients from the experimental data on which their conclusions are based. If interpreted correctly, the models are used to construct a kind of theoretical 'behavioural' landscape depending varying boundary conditions and damping coefficients, after which it could be established where the experiments map (how? optimization modelling?) on that landscape (but the question remains why then return to the regressions?). Most probably, this remark originates from my too limited knowledge of, and skills in, maths. Since I cannot believe the maths are there just to overwhelm the maths-layman, (but interested biologist, physiologist, etc.), the line of thought should be much better explained (preferable in a more colloquial, verbal way, moving the equation deduction to an appendix or to supplementary material). On the other hand, if my initial impression, that the extensive presentation of the mathematical models is, to a large extent, redundant for the main message, is correct, why present these anyway?

Thank you for your appreciation of our paper. Both the other Reviewer and the Editor agree with you in challenging the relevance, in the very core of the main text, of a long and articulated mathematical description of the models and their application to extract damping coefficients from experiments. By following your request, we moved the integral mathematical part to Supplementary Material and left a summarized process of just 13 equations (out of 47), presented in more colloquial way. Within the main text we maintained the original equation numbering to allow readers to easily find their exact location in Supplementary Material.

In addition, it is true that in one of the investigated conditions (the swinging straight pendulum) the 'simple' exponential regression of oscillation (rectified) peaks allows to obtain the damping coefficients, but there are reasons to show the extended solution (eq. 23) for the angle time course. The first one is that next investigators on this subject might be interested in representing in the phase plane both the experimental spiral-like data (see fig. 2) and the theoretical continuous trajectory, in order to better distinguish between 'truly passive' performances and the ones affected by some muscle activations. Although the case of finding a close form solution of straight damped pendulum is a 'simple' classic challenge for mechanical engineering students, there are specificities of our pendulum that make the solution not easily available from the Internet. The second reason is that the conclusion about the 'simple' analysis of oscillation peaks in straight damped pendulum wouldn't have been reached prior to inspecting the final equation (eq. 22 o eq. 23)

As far as the inverted damped pendulum (in the unload and loaded conditions) is concerned, we have tried to explain that there is no 'easy processing' of the experimental data, particularly due to the 'one only shot' per trial, with a quite limited angle span involved. Here, to allow experiment/processing reproducibility in the future, we need to provide as much details as possible.

Appendix B

Reply to reviewers (*in Italic*)

Associate Editor Board Member

Comments to Author:

Thank you for submitting your revised manuscript to Proceedings B. We appreciate the effort that authors have put into revising their paper, and we note that the number of equations within the main body of text has now dropped significantly. The referee has only a single comment/question regarding the actual content of the manuscript. Otherwise, their outstanding concern still remains the 'readability' of the manuscript. Whilst we recognize this has already improved considerably, please take note of their suggestions with regard to the structure and narrative of the methods.

We moved the equations and related text on supplementary materials. Now there are only 8 equations in the main text. Material and Methods have been improved accordingly to the reviewer's suggestions by moving at the beginning the experimental protocol and on supplementary material the computational part of b.

Reviewer(s)' Comments to Author:

Referee: 2

Comments to the Author(s).

Dear Editor,

I have considered the revision of the paper by Minetti et al. on 'Frictional internal work of damped limb oscillations'. I'm still convinced about the importance of the results presented in this study, which is very well framed in the introduction. The presented findings may be relevant for proximate physiological, functional morphological, kinesiological and biomechanical studies but, potentially also, for instance, for the proper understanding of the evolution of locomotor behaviour. Therefore, to attract as many readers as possible (and to convince them immediately about the main message: 'frictional work cannot be neglected'), I would personally still prefer to keep the body of the paper abstract of maths as much as possible. This does not mean I doubt its relevance, but I'm convinced that it will be beneficial for the readability of the paper if most maths are moved to an appendix or supplementary material. I don't see, for instance, what the benefit is of having equations 23 or 33-35 in the text.

DONE, we moved these equations and related text in the supplementary materials. Now the equations are just 8 instead of 46 of the first submission.

It seems more important to me to mention clearly what is aimed at with the experiments and models (something like "Knowing the initial pendular state (angular position and speed) and based on the morphometrics of the (straight and inverted) pendulum (inertial properties) it is possible to deduce the damping coefficient that reproduce the experimental oscillations or limb swings (for the mathematical details, see appendix or supplementary material) in order to maximize focus on the important results.

It is very difficult to synthesize in a single sentence the common features of data analysis of unloaded straight and loaded-unloaded inverted pendula, actually because the two conditions

(despite their common mechanic paradigm) are too different. We added, just before 'Statistical Procedure' (Lines 200-210) the following sentence:

'Finally, the data analysis leading to the relevant damping coefficients in this investigation can be summarized as: Straight Pendulum – after checking for passive oscillations the angular peak decay in successive swings was processed; Inverted Pendulum - by knowing the initial and final pendular state and based on its inertial properties when unloaded or loaded pendulum it is possible to deduce the damping coefficient that reproduces the experimental oscillations or limb swings.'

Similarly, while definitely relevant, the discourse on what the appropriate regression procedure is (L205-230) is, according to me redundant for the body of the paper (but again, relevant for an appendix). This is, of course a personal view and preference.

Although the authors agree about the unorthodox presence of a statistical issue taking a big space in relation to its apparent relevance, they feel that it is important to retain it in the main text. This little story mines the acquired confidence in statistics, and particularly in a very diffuse tool as Least Squares Method. To one of the authors' knowledge, who undertook a 4 years Specialty degree in Statistics, this potential oversight is never addressed by the Course Syllabus. Also, there are topics that are too tiny to be published in technical or methodological journals and often find their space only when quoted as relevant in research papers. In synthesis, by relocating this paragraph in Supplementary Materials would likely spoil, throughout a lesser visibility, an opportunity for readers to better discern a worldwide applied method, here crucially related to a part of data analysis.

In its present format, I would also suggest to move the description of the experimental protocol to the beginning of Materials and Methods. The reason for this is that much of the theoretical explanations become more meaningful when the reader has already an idea of the experiments performed. This is especially true for what is written in L178-185. It's difficult to understand what is meant if one has no idea about what the 'inverted' experiments involve.

DONE, we moved the 'Experimental Protocol' at the beginning of Material and Methods as suggested.

There is also one thing that puzzles me. It concerns what is mentioned in L185-190. I'm confused about what is written here. I interpret this as if multiple damping coefficients can be selected by the algorithm that can all provide a proper representation of individual experimental results. According to me, there can be only one beta that results in the correct angular position and speed after time t-swing.

For each goal, i.e. reaching a given final angle, final angular speed and a total swing time as close as possible to the experimental data, threshold were manually increased as to obtain 3 b values (for each experimental trajectory) whose coefficient of variation was smaller than 1%. The first set of (increased) thresholds fulfilling this requirement was taken and the mean value of the 3 b charted. The aim was to minimize b estimate heterogeneity in the same experimental inverted swing. There was also a typo ('where' instead of 'were'). Now the sentence (moved on Supplementary Materials) reads 'Thresholds for the allowed approximation of the three simultaneous goals were manually increased as to obtain a coefficient of variation of... '.